# Microbeam X-ray Reanalysis on Periodically Assembled Poly(β-Hydroxybutyric acid-Co-β-hydroxyvaleric acid) Tailored with Diluents

**DOI:** 10.3390/polym15163484

**Published:** 2023-08-20

**Authors:** Chun-Ning Wu, Selvaraj Nagarajan, Li-Ting Lee, Chean-Cheng Su, Eamor M. Woo

**Affiliations:** 1Department of Chemical Engineering, National Cheng Kung University, No. 1, University Road, Tainan 701, Taiwan; etning@gmail.com; 2Department of Materials Science and Engineering, Feng Chia University, Taichung 407, Taiwan; ltlee@fcu.edu.tw; 3Department of Chemical and Materials Engineering, National University of Kaohsiung, No. 700, Kaohsiung University Rd., Nan-Tzu Dist., Kaohsiung 811, Taiwan; ccsu@nuk.edu.tw

**Keywords:** poly(β-hydroxybutyric acid-co-b-hydroxyvaleric acid), diluents, periodic self-assembly, synchrotron X-ray diffraction

## Abstract

Self-assembly of 3D interiors and iridescence properties of poly(β-hydroxybutyric acid-co-β-hydroxyvaleric acid) (PHBV) periodic crystals are examined using microcopy techniques and microbeam X-ray diffraction. Morphology of PHBV can be tailored by crystallizing in presence of poly(vinyl acetate) (PVAc) or poly(trimethylene adipate) (PTA) for displaying desired periodicity patterns. The regular alternate-layered lamellae of banded PHBV crystal aggregates, resembling the structures the natural mineral moonstone or nacre, are examined to elaborate the origin of light interference and formation mechanisms of periodic lamellar aggregation of PHBV spherulites. By using PHBV as a convenient model and the crystal diffraction data, this continuing work demonstrates unique methodology for effectively studying the periodic assembly in widely varying polymers with similar aggregates. Grating structures in periodically assembled polymer crystals can be tailored for microstructure with orderly periodicity.

## 1. Introduction

When crystallized from the melt, molecular chains of semicrystalline polymers arrange themselves, via crystal-by-crystal, into aggregated spherulites, and occasionally these spherulites may display orderly patterns of periodicity [1,2,3,4,5]. One of the most prevalent studies investigates the mechanisms of periodically assembled spherulites, in which the polycrystalline aggregates display fascinating alternate-layered crystal arrangements that self-repeat in birefringence patterns displaying vivid optical rings. Periodic-aggregated spherulites have been reported in many polymers, more commonly seen in high-density polyethylene (HDPE) [6,7,8,9] and aliphatic and arylate polyesters [10,11], bioresourceful polyesters such as poly(ε-caprolactone) (PCL) [12]. Poly(p-dioxanone) (PPDO) can also be packed into crystals with distinct periodic birefringence patterns [13,14,15]. Most of the conventional and classical studies on periodic banded spherulites frequently assumed propositions that the polymer spherulites are packed with pseudo-two-dimensional, giant screw-like single-crystal lamellae with periodic pitches matching with the optical band spacing, and that continuous helix-twist is a result of chain-folding-induced stresses on lamellar crystal plates [16]. Yet, it should be cautioned that in films of finite thickness of 5–20 µm showing banded morphologies, the crystal growth and assembly filling the space is 3D. Thus, assumption of 2D planar assembly of a single-crystal-like lamella with synchronizing helix-twist to produce periodic bands may be improper or somehow inadequate with hidden fallacies. Secondly, direct morphological evidence for giant and continuous helix-twist screw-like lamellae (as long as 30–500 µm from nucleus to periphery rim) has not been proven in numerous studies conducted during many past many decades. Typically, there are only sporadic reports about localized and random lamellar twists in etched specimens whose helix-screw pitch never matches with the optical interband spacing [17,18]. The periodicity in poly(ethylene adipate) (PEA) crystals was expounded through a more realistic three-dimensional (3D) interior approach coupled with in-depth analysis on the lamellar arrangements on top-surface relief patterns and fractured interiors. Later, similar novel approaches have also been utilized to better understand the interior assembly of classical polyethylene (PE).

A recent review article [19] on banded polymer spherulites has critically pointed out that the dramatic mismatches of optical interband spacing with the actual helix pitches of “lamellar helices”, as evidenced by SEM/TEM microscopy data from many past decades, have remained a gap that scarcely bridges the classical models of helix twist to direct experimental morphological evidence. The optical interband spacing of PE ring bands are known to be ~2–4 µm (depending on T_c_ values) according to the POM evidence; many dedicated investigators have searched for evidence of lamellae helices during the last 60 years to justify the presence of “helices” in banded polymer spherulites. Keller and Sawada did earlier pioneer work in 1964 [17] to investigate PE, which has an optical inter-ring spacing ca. 4 µm, but a lamella helix pitch by using TEM is ca. ~1 µm. That is, these two results are mutually not in agreement [17], with discrepancy between the optical ring spacing and actual lamella helix pitch being as large as four times. In 1995, Kunz et. al. [18] presented the morphology of screw-like PE lamellae (crystallized from a gel medium) using TEM showing a “helix” pitch to be ca. ~100 nm [18], which disagreed with results indicating a 1 µm lamellar helix reported earlier by Keller [17]. In addition, Kunz et al.’s earlier work [18] also led to disagreement regarding optical ring spacing with the lamella pitch (disclosed in TEM evidence) in the order of magnitude difference by ca. 40 times. Similar mismatches of the helix pitches and morphological evidence for “helices” are widely seen in the banded spherulites of other polymers, such as PLLA, PHB, PHBV, etc., as surveyed in a review article [19].

In recent studies on PHBV periodic crystal assembly, i.e., ring-banded spherulites (RBS) [20,21,22,23], the possible influence of specimen film thickness on internal periodic patterns was thoroughly investigated to prove that PHBV spherulites crystallized from PHBV/PVAc (70/30) blend, in films of ~1 µm, showing no periodic bands but only straight or curved dendrites in the final crystal aggregates. In contrast to the regular thin film thickness (~10 µm) to bulkier thickness (~50 µm), the crystallized pattern would show a double-ring-banded morphology. This evidence demonstrates that periodic assembly requires a minimum film thickness, below which polymers cannot self-assemble into periodic aggregates. This indicates that chain-folding and induced stress similarly occurs in thin or thick films as long as the thickness of specimens is greater than the thickness of a typical single lamella (~10–15 nm), but periodicity in the final high-hierarchical aggregates may disappear when the film thickness is smaller than 1–5 μm. This is also indirect yet critical proof that the proposition of chain folding-stresses as a main accounting factor of lamellar twist and banding phenomenon may deserve reassessment. If chain-folding stresses were factually responsible, then stress-induced helix-twist and periodic banding will always be present, even in single-crystal lamellae with film thicknesses as thin as 10–15 nm, and thus periodic optical bands will be observable in both thin and thick films, which contradicts the experimental facts. In addition to ultra-thin-film PHBV proven to show no periodic bands [24], other polymers in confined thin films behave similarly [25,26,27,28]. According to the classical Aristotle’s proof-by-contradiction, the chain-folding-induced proposition apparently does not stand to the test by the experimental fact that all polymer films (e.g., PHBV) with ultra-thin-film thickness (<1 µm) do not display periodic bands [21].

These long-standing discrepancies further strengthen and confirm that alternative novel approaches, using innovative interior dissection and microbeam X-ray characterization, are timely and critical to disclose more plausible mechanisms. In addition, the grating assembly actually has potentials for interfering with white light to perform iridescence functions, similar to those seen in many biospecies or minerals generated by biospecies. By interior dissection analyses, PHBV has been shown to exhibit well-defined large banded spherulites in a large range of temperature and within a range of film thickness [20,24]. The search for scientific origins and mechanisms in polymer systems has been a continuing work and extensively conducted for many decades. Blending semicrystalline polymers with amorphous diluents (molten/amorphous polymers or compounds) is one of the attempts to promote the formation of banded spherulites and to elucidate the structure crystalline portion of banded spherulite after the removal of the diluent via solvent etching [20,24]. In 2013, Woo et. al. conducted critical research on the 3D lamellar assemblies of PHBV ring-banded spherulites crystallized from a blend with amorphous diluents such as poly(vinyl acetate) (PVAc) [24], and reported fine interior crystal architectures leading to periodically optical birefringence bands. This on-going work further refined and extended a previous microbeam X-ray analysis [28]. Together with a reinterpretation of the microbeam X-ray data, the controversial issues of self-assembly of PHBV were addressed by providing more sophisticated 3D dissected morphological data to reinvestigate several critical aspects.

## 2. Experimental

### 2.1. Materials and Preparation

PHBV (HV = 8–11 wt.%) was obtained from Aldrich (St. Louis, MO, USA), which has T_g_ = −10 °C and T_m_ = 146 °C. The M_w_ and PDI are 39,000 g/mol and 1.48, respectively. The chemical structure of PHBV is given in Figure 1. Two polymeric diluents were used for modulating the most suited morphology in PHBV. Due to its phase miscibility with PHBV, low-crystalline poly(trimethylene adipate) (PTA) was used to modulate the banding patterns in crystallized PHBV specimens. PTA was obtained from Scientific Polymer Products, Inc. (Ontario, NY, USA), with M_w_ = 8900 g/mol, PDI = 1.28, T_g_ = −63 °C, and T_m_ = 38 °C. Amorphous poly(vinyl acetate) (PVAc) was also used, which was obtained from Fluka, Inc., with M_w_ = 260,000 g/mol and T_g_ = 34 °C. PVAc was used as a codiluent to blend with PHBV for examining the universality of PHBV architectures in ring-banded assembly.

PHBV/PTA or PHBV/PVAc blends were well mixed using chloroform (CHCl_3_) as a common good solvent with 4 wt.% concentration and the composition fixed at 80/20 (wt./wt.) for PHBV/PTA or PHBV/PVAc. Films were prepared by drop-casting on glass substrates and dried-off solvent at 40 °C. The sample films were prepared in two nominal thickness levels: (normal ~ 10–20 µm for POM/AFM and thick ~ 20–50 µm for SEM). Specimens were premelted on a hot plate to a maximum melt temperature (T_max_ = 220 °C) for two minutes, then quickly replaced to another hot stage set at specified T_c’_s (T_c_ = 100 to 60 °C) until full crystallization. This rapid transfer operation was aimed to quickly equilibrate the molten specimens at a designated isothermal temperature. Acetone, being an effective solvent for dissolving PTA and PVAc but not PHBV, enables the convenient partial removal of the PTA and PVAc constituent from the crystallized specimens, although complete elimination is not achieved.

### 2.2. Apparatus

Atomic-force microscopy (AFM, diCaliber, Veeco Corp., Santa Barbara, CA, USA) was used with a silicon tip (f_o_ = 70 kHz, r = 10 nm). Both AFM height profiles and phase images were also scanned on film samples.

A polarized-light optical microscope (POM, Nikon Optiphot-2, Tokyo, Japan) was equipped with a Nikon Digital Sight (DS)-U1 camera control system and a microscopic hot stage (Linkam THMS-600 with T95 temperature programmer). A tint λ-plate was used for birefringence contrast colors.

Field-emission scanning electron microscopy (SEM) was performed using a Hitachi-SU8010 (Hitachi, Tokyo, Japan) microscope. HR-FESEM was utilized for characterizing the fracture and top free surface. Standard platinum sputtering (10 mA, 300 s) was applied to the specimens prior to SEM characterization. To expose the inner lamellar skeletons, specimens of crystallized polymer films were fractured and precut on a glass slide using a diamond knife. As a fracture propagated statistically across a number of spherulites, experience was used to select specific portions for best results. In the SEM chamber, the fractured samples were slightly tilted and carefully positioned so that the focus of the electron beam could be placed simultaneously on both the top and fractured interior surfaces of specimens.

## 3. Results and Discussion

### 3.1. Periodic Assembly: Top Surface vs. Interior

As melt treatments of polymers at different T_max_ might influence the growth kinetic and thus their band patterns. AFM characterization was conducted on top surfaces to reveal the detailed lamellar patterns of the pristine and acetone-etched ring-banded PHBV spherulites (crystallized in PHBV/PTA (75/25) blend). Figure 1 exemplifies (A) AFM height image of an entire PHBV spherulite, and respective zoom-in images to regions 1, 2, and 3, as marked on images. For etched specimens, Figure 1A is the AFM height image of a pristine (unetched) PHBV spherulite, with (1, 2, 3) being zoom-in phase images to respective blocks (1, 2, 3). The nucleus center is packed as a ball-like domain (d = ~10 µm). From the center, successive rings surround it and continue until impingement. One can see clearly that on the ridge bands, all fibrous crystals are aligned in the radial direction, initially linearly. Yet, toward the lamellar tails, all cilia crystals bend in the circumferential direction and twist to merge into a valley band (dark crevices in AFM images for zoom-in blocks 1, 2, 3). The interband spacing is ca. 7–8 µm on average (midline of a ridge to that of the next ridge). Figure 1B shows the counterparts of solvent-etched specimens. Similar features as those in PHBV subjected to T_max_ = 190 °C are observed in this PHBV specimen subjected to T_max_ = 220 °C. The spacing is ca. 7–8 µm on average (midline of a ridge to that of next ridge). The AFM height image shows that the valley crevices increase in width of gap after acetone etching, but the interband spacing increases. This is quite expected as the etching had removed the amorphous components accumulated in the valley region sandwiched between successive ridge bands. The spacing remains the same at 7–8 μm, indicating that etching removed some of the amorphous (less crystalline) components but did not alter the band morphology. Solvent-etching tends to remove amorphous cover layers from the up-and-down waving ridge and valley; thus, the vertical height drop from ridge to valley of the bands is ca. 450 nm for the etched samples, which is 200–300 nm greater than the nonetched (pristine) samples.

Solvent-etching helps to expose the inner architecture hidden below the top layers. In the solvent-etched PHBV specimens, it is clearer that the PHBV banded spherulites are composed by radially linear lamellae arranged in a radial direction on the ridge band, and bent/twist lamellae are arranged in the tangential direction on the valley band (i.e., interband crevices). The growth cycles are repetitive and periodic in the same fashion. On the ridge bands, all fibrous/cilia crystals are self-aligned in the radial direction, and toward the tails, all cilia crystals bend in the circumferential (tangential) direction and twist to merge into a valley band (i.e., the dark crevices in AFM images for regions 1, 2, 3). Additionally, the end of these finer/thinner lamellae on the ridge turn downward to the valley into a slightly bulkier bundle arranged in a clockwise circumferential direction (tangential direction), where they turn to the next ridge, and the lamellae revert back to the original finer crystal lamellae arranged in the radial direction. Cycles of growth and assembly repeat in same manner.

AFM analysis was performed on a PHBV specimen (acetone-etched) subjected to a higher T_max_ = 220 °C but crystallized at same T_c_ = 80 °C, and the results are shown in Figure 2. Generally, the morphology is similar for PHBV specimens subjected to T_max_ = 220 °C or T_max_ = 190 °C, but the fibrous cilia lamellae are coarser in the PHBV banded spherulites subjected to T_max_ = 220 °C. That is, the fibrous lamellae in the PHBV banded spherulites treated at lower T_max_ = 190 °C are statistically finer and thinner than those of the PHBV ones treated at higher T_max_ = 220 °C. In addition, upon crystallization at the same T_c_ = 80 °C, the interband spacing (δ = 16 µm) in the PHBV spherulites subjected to T_max_ = 220 °C is almost twice as wide as and larger than those in PHBV subjected to T_max_ = 190 °C (δ = ca. 8 µm). Nevertheless, the general feature of periodically assembled morphology, with the radial-oriented lamellae bending from the ridge region (bright band in AFM image) to 90° angle to merge into the valley region (dark band in AFM), is similar regardless of the melt treatments at T_max_ = 190 or 220 °C. Apparently, the periodic ring patterns, lamellar assembly, and morphology of PHBV are highly influenced by kinetic factors (T_max_ or T_c_) imposed during crystallization. Distinct discontinuity (marked with blue arrows) is present between successive cycles of rings, suggesting that the packing growth is clearly not composed of continuous helix-twist lamellar plates, but intermittent growth-precipitating termination occurs at the valley bands by tail-thinning to fibrous tips (marked on the images by blue arrows). This does not mean that lamellar plates do not twist; the data suggest that they are not continuous. Lamellar assembly in etched vs. unetched (pristine) specimens was analyzed to elucidate the top-surface assembly. Solvent etching serves two purposes: (1) expose the interlamellar cracks and (2) enhance the morphological contrast of perpendicularly intersecting lamellae. Figure 2B is the tapping-phase images of lamellar assembly in PHBV spherulites: (a) unetched vs. (b) solvent-etched specimens. Discontinuity, exemplified by radial-oriented crevices in between the interfibril bundles, is also seen. The crevice discontinuity (radial-oriented) suggests that the lamellae constituting the ridge bands are all radial-oriented, which are therefore more easily detached upon solvent-etching.

The lamellae, on the top surface as well as in the interior of the banded PHBV spherulites, self-assemble in the same repetitive manner to form successive bands until impingement, and these lamellae on the successive ridges are interconnected by clockwise bending and circumferentially arranged lamellae on the valley. For comparison with AFM analysis, selective SEM characterization was performed on the same banded PHBV spherulites. Figure 3 displays (a) SEM micrographs and (b) a schematic diagram for assemblies on the top surface that correlate with fractured interior lamellae of PHBV crystallized from the PHBV/PTA (75/25) blend at T_c_ = 80 °C.

Assembly of lamellae beneath the top surface showing an alternate up-and-down topology could be cracked open only with a 3D view into the interior. Although PHBV diluted by PTA was used for better viewing the discontinuity between interlamellar bundles, its morphology, however, might be slightly adjusted by presence of the diluent. Thus, a specimen based on neat PHBV was similarly analyzed using the 3D dissection technique. Figure 4a shows an SEM image of the interior dissection of neat PHBV periodic bands crystallized at T_c_ = 80 °C. In the interior, the valley–ridge up-and-down topology is no longer discernible; instead, periodicity and growth cycles of alternate tangential- and radial-oriented lamellae (correspondingly: zone-II = normal oriented and zone-III = horizontal-oriented lamellae), with a sandwiched zone-I clearly visible. For better appreciation of the interior lamellae showing alternate normal/horizontal transition along the radial direction, a simplified scheme is illustrated in Figure 4b, where V = valley, R = ridge on the top surface. As depicted in Figure 4b, a straightforward correlation is seen in which the tangential-oriented lamellae are situated directly beneath the ridge (R) band on the top surface, while the radial-oriented bands are situated directly beneath the valley (V) band.

The fact that there are cracks or crevices existing between the successive bands and interior layers of lamellar bundles provides evidence of periodic discontinuity, due to fractal-growth cycles in the lamellar assembly, and indicates that the lamellae do not twist continuously into helicoids like DNA molecules. In addition, orientation of the cracks and their locations on the ring-banded spherulites support the interpretation of the lamellae assembly on the top surfaces vs. interiors of PHBV. The cracks always appear after solvent-etching and are never seen in pristine (unetched) PHBV samples, indicating that cleavage between lamellae and crystal deformation (bending, scrolling, twisting) is induced by stresses of solvent exposure. These results indicate that the layered bands on both the top surface and cross-section are discontinuous. Through a simple schematic and analysis of orientation of crevices along the lamellar orientations, the fine architectural arrangements on the top surfaces and interiors can be better understood.

For a more universal proof, the morphological result of self-assembly of poly(3-hydroxy butyrate) (PHB), with a chemical structure similar to PHBV (a copolymer of PHB with HV = ca. 8–11%), is used as a demonstration. Neat PHB, when crystallized at T_c_ = 80 °C, displays similar ring bands as does PHBV; however, the patterns in neat PHB are much less orderly. An earlier study [29] has shown that discontinuity does exist in the seemingly continuous banded patterns of PHB; i.e., the lamellae self-assembled in the radial direction are not really continuous uninterruptedly from nucleus to periphery of circular/spherical aggregates, but periodically grating-like with periodic interfaces between the successive onion-like layers of alternate tangential direction and radial direction. Figure 5a,b demonstrate clearly that circumferential discontinuity, coinciding with the ridge/valley banding pattern as cracks induced by rapid liquid N_2_ cooling or solvent etching, exists in the ring-banded PHB crystallized at T_c_ = 80 °C (after subjected to T_max_ = 210 °C for 4 min). Once again, discontinuity, existing as intersection of two perpendicularly oriented lamellae, appears to be universal and not an incidental happenstance. Obviously, the lamellae in ring-banded spherulites may bend and twist locally in transiting from a ridge to valley band, but they are not continuous as are DNA molecular helices. This assembly in PHB is similar to that in PHBV discussed in previous figures. Universality of periodic assembly with discontinuity between the radial-oriented and tangential lamellae is proved in neat PHB as well as PHBV.

The assemblies on the top surface and interiors are now clear enough for PHBV. To describe in detail the 3D architectures is now possible. Graphical schematics for 3D lamellar assembly in the ring-banded PHBV spherulites are further shown and discussed. Figure 6A depicts the top-surface bands with periodic discontinuity, where the interior assembly would have been masked if dissection were not performed. Figure 6B illustrates the interiors with the top-surface banding removed for clarity of the lamellae therein, where the ridge and valley alternate in periodical packing to form the final periodically ring-banded PHBV crystal aggregates. Figure 6C is the 3D view representing the architecture on both the surface relief and interior of lamellae assembly in the banded PHBV spherulites that had been solvent-etched to expose the interior interbundle crevices. In the ridge-band region (top surface and interiors), the lamellae grow perpendicularly to the substrate; however, due to the space constraints on the top surface, the emerged lamellae change their growth directions to the radial direction upon reaching the valley. In the valley band (top surface and interior), the lamellae bend at about a 90-degree angle to the tangential direction covering the underside lamellae of the valley before they collide with the lamellae beneath the next ridge and continually grow in the radial direction.

From the 3D interior analyses, the lamellae of the ridge slowly change in direction by bending and/or branching to the lamellae of the valley, then again by bending and branching slowly change their growth direction to the lamellae of the ridge, forming a concave U-shape architecture. These sequences in the interior lamellae repeat in periodic cycles finally to form a 3D ring-banded spherulite. On the top surface of the banded PHBV spherulite, the periodic bands are either concentrical circles or spirals around a nucleus region with a diameter of ca. 5–10 μm. Note in these schemes, the nucleus region is truncated from the scheme for brevity of discussion. For the outer portion near the nucleus of spherulites, the lamellar assembly in the nucleus eye is more complex where relatively irregular arrangement of lamellae is usually seen. Regularity of periodic bands usually starts in the regions outside the nucleus core.

### 3.2. Microbeam WAXD Analysis for Lamellae in Ring-Banded PHBV

Figure 7 shows schemes of two different models: the corrugated-board assembly with mutually perpendicularly oriented lamellae accounting for the ring bands vs. the classical model of continuous helix-twist lamellae. From the schemes shown in Figure 7a,b, it appears that the crystal units of the assembled lamellae in these two models both rotate and tilt; however, they may rotate in certain different ways. Note that according to the model of continuous screw-like twist, the crystal units not only tilt but also spin–rotate as the lamellae undergo a twist; by contrast, according to the corrugated-board assembly, the crystal units mostly tilt to different oblique angles as the microbeam is moved from f14 (centerline of a valley) to f18 (centerline of next valley). First, we assume that both models are just propositions for interpreting the periodic birefringent bands in crystallized PHBV. Initially, however, they must stand to experimental tests by SEM morphological and microbeam X-ray data. Interior dissection by SEM characterization has been discussed in previous morphological results, proving that assembly of cross-hatch architecture is confirmed by SEM interior dissection. Next, we focus on microbeam X-ray data, where the X-ray microbeams are to be step-moved along a radial direction across successive ridge and valley bands, as illustrated in Figure 7(a1) and Figure 7(b1), respectively. Validity of these two datasets is discussed in the figures that follow.

Microbeam WAXD data were reanalyzed to prove the microscale localized assembly of lamellae in periodic cycles in banded PHBV spherulites. Note that PHBV ring-banded spherulites crystallized from PHBV/PTA or PHBV/PVAc yielded a similar morphology. Figure 8a,a1 present a POM graph for PHBV banded spherulite crystallized from PHBV/PVAc (70/30) at 110 °C, and the enlarged micrograph reveals microbeam WAXD spots step-moved from spots f14 to f18. Figure 8b–d are the interior-lamellae morphology of PHBV-RBS with interband spacing ca. ~10 μm as viewed in an SEM micrograph, unit cells with orientations of crystal planes, and 2D-WAXD results, respectively. The interband spacing from valley to valley is covered with several microbeam WAXD spots f14–f18. Spot f14 is located on the valley, where signals of (110) and (020) crystal planes appear when the c-axis is parallel (flat-on lamellae) to the microbeam axis. It is interesting to note that the diffraction result indicates the possibility of two opposite crystal alignments (Figure 8d), and diffraction results indicate two different crystal alignments in RBS. Importantly, the SEM morphology reveals that the horizontal lamellae are packed beneath the valley region, which is in agreement with the WAXD pattern. Spot f15 lies on inclined lamellae (not right-normal but at oblique angles with respect to the substrate) near the valley–ridge transition zone, so a weak signal of (020) crystal plane is observed, which is in agreement with the crystal cell’s orientation. Spot f16 is positioned just beneath the ridge zone, so the signal for the (020) crystal plane disappears and the appearance of the signal for the 101/111 crystal planes confirms the vertical (normal) lamellae’s orientation with respect to the substrate. These vertical-oriented lamellae in the ridge zone gradually tilt and finally turn to horizontal lamellae in the valley zone at spots f17 and f18, which resemble the signals for spots f14 and f15 in the previous cycle. That is, a cycle of growth is completed when the microbeam is moved from spots f14 to f18. Further, according to the SEM morphology for PHBV, the grating cross-bar structure consists of a discontinuous assembly composed of alternately vertical and horizontal lamellae, suggesting that the SEM morphology and microbeam WAXD analysis are in good agreement with the grating lamellae assembly.

To ensure the reproducibility of the morphological results, the interior surface of an alternative specimen with a thicker film of ca. ~25 μm PHBV crystallized from PHBV/PVAc (70/30) banded spherulite at same T_c_ is shown in Figure 9. The SEM evidence clearly shows that the vertical lamellae under the ridge intersect in a cross-hatch fashion with the horizontal lamellae beneath the valley. When these horizontal lamellae reach a certain length, the crystals bend or reoriented vertically to form a ridge zone. This result demonstrates that the film thickness does not influence the cross-hatch grating assembly of PHBV/PVAc banded spherulites. There is no experimental SEM evidence for claiming continuous helix-twist lamellae in the interiors of the banded PHBV. Fully grown lamellae can self-nucleate in each cycle of growth to spawn new lamellae to grow discretely, and the new lamellae self-orient in alternate directions to complete cycles of repetitive periods in the same manner. The alternate vertical lamellae in the ridge, or oblique-oriented lamellae in the transition zone, and horizontal lamellae in the valley zone complete a cycle of assembly, which is responsible for generating optical birefringence-patterned rings in the PHBV spherulites when crystallized at a suitable range of T_c_.

Classical Photo-51 for proof of DNA helices is used here to test for or against the proposition that the crystal plates exhibit a similar continuous helix-twist as DNA molecules, which essentially are “polymer chains, but not a lamella”. To test the validity of an assumed helix-twist crystal plate in the banded PHBV aggregate, with lamellar helix-twist conformation like that in DNA molecules showing a classical Photo-51 X-ray diffraction pattern, actual experimental X-ray patterns for banded PHBV are described, which can then be judged with Aristotle’s proof-by-contradiction. Figure 10 displays an expected X-ray diffraction 2D pattern on eight different spots traversing one pitch of a supposedly helix-twist lamella, with typical dimensions of 50–100 μm in the axial length, 2–3 μm in gyro-diameter, and 5–10 μm in pitch. The X-shaped diffraction pattern emerges from the edge-on lamella with azimuthal deviation in spots 1 and 2. However, spots 3 to 6 are both located in a flat-on lamella, so there would be no diffraction pattern to be seen. Microbeam diffraction at spots 7 and 8 leads to the reappearance of the X-shape diffraction pattern. So, there is no experimental result in PHBV-RBS to support this kind of X-shape pattern in X-ray diffraction. Note here that the classical Photo-51 for proof of DNA molecular helices is used as a model for testing the proposition. Thompson et al. [30] used the X-shape pattern in X-ray diffraction of DNA molecules to prove that Rosalind Franklin’s classical X-ray photo (known as Photo 51) corresponds to a helical continuous bend if an X-ray beam is moved along the long axis of the helix screw. Atkins and de Paula [31] described that a single turn of the helicoid (double helices held by numerous bases units) of the DNA molecules present as two planes intersecting at an artificial azimuthal angle, and these two planes are oriented with an angle of +α° to the horizontal line and another at angle −α°. Therefore, the X-ray diffraction pattern (an X-shape) can directly measure the helicoid angle (±α°) [31]. If a single lamella in banded PHBV indeed rotates like a screw (a helicoid) and an X-ray microbeam is moved along the axis of the helicoid, this situation is similar to an earlier experiment described by Atkins and de Paula [31], which, however, is not observed in typical microbeam X-ray characterization on banded PHBV in this work, or banded PE [32], PEA [33], or banded PTT [34,35,36] reported in the literature.

## 4. Conclusions

Critical breakthrough novel findings in this work can be summarized. By using the model of PHBV’s self-assembly into periodically banded aggregates (crystallized from PHBV/PTA or PHBV/PVAc), this work further extends that the repetitive morphology is a hierarchical crystal complex composed of fractal branches that are alternately normal- and radial-oriented during growth. In corresponding to the POM patterns, they appear as blue/orange optical bands. Such crystal assembly closely resembles those seen in nature’s numerous organic/inorganic species possessing periodic crystal gratings for producing colorful iridescence spectra by structural interference with white light. Dissection of interiors by SEM evidence clearly exposes that the cross-bar pitch exactly matches with the interband spacing as viewed in the POM results. Periodicity in growth is initiated immediately from the two ends of elongated crystal sheaves in the central nuclei. For a banded crystal aggregate on a flat substrate, the lamellae beneath the ridge zone self-assemble vertically to the substrate, then reorient horizontally beneath the valley band. Growth cycles repeat in the same manner repetitively until drainage or impingement. The reason for the periodic up-and-down topology is not due to helix screw’s groove, but rather it is attributed to periodic alteration where the radial-oriented crystals beneath the valley extend upward to reach the surface; subsequently, they bend and become submerged to reoriented into horizontal lamellae in the valley band, forming a “concave-up” bending.

Obviously, the sophisticated interior dissection of the banded PHBV spherulites clearly displays the interplay between the crystalline lamellae of hierarchical levels (main and side-branches), which results in mutually perpendicular intersection of two crystal species whose interior pitch matches correspondingly to the interband optical spacing. By reinterpreting the previous synchrotron microbeam X-ray data, the microfeatures of the periodic grating architectures are further substantiated. The proof of 3D morphological results and microbeam 2D WAXD analysis further reinforces that the assembly in the PHBV banded periodic architectures are that the alternate optical birefringence bands are mainly accountable by the grating assembly. The alternate grating periodicity in the banded PHBV can be a mimicry of many nature’s structural crystals to function as photonic units for interference.

## Data Availability

Data are contained within the article and are available upon reasonable request.

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
