# Peer review of "Microbeam X-ray Reanalysis on Periodically Assembled Poly(β-Hydroxybutyric acid-Co-β-hydroxyvaleric acid) Tailored with Diluents"

_polymers, 2023, doi:10.3390/polym15163484_

Round 1

Reviewer 1 Report

The authors submitted a manuscript titled “Microbeam X-ray Re-analysis on Periodically Assembled Poly(β-hydroxybutyric acid-co-β-hydroxyvaleric acid) Tailored with Diluents” in which they examine lamellar shapes of poly(b-hydroxybutyric acid-co-b-hydroxyvaleric acid) blended with poly(1,3-trimethyulene adipate) (PHBV/PTA) based on data derived by atomic force microscopy and scanning electron microscopy. The data are then used to conclude on the nature of lamellar twisting in this group of polymers. In a second step they compare the derived morphological information with X-ray micro diffraction data from a similar material (poly(3-hydroybutyrate-co-3-hydroxyvalerate) blended with poly(vinyl acetate) as dilutend, (PHBV/PAVc), published by Hsieh Y-T et al. prior in POLYMER 55 (2014) 6906-6914.

First of all I do not think that the title of the manuscript describes the content of manuscript adequately, since the title implies that the main focus of the manuscript focusses on the reanalysis of X-ray micro diffraction data, which then are used to describe the banding pattern in melt crystallized spherulites of (PHBV/PTA). However, the majority of the manuscript describes the detailed analysis of morphological features and the conclusions derived from them. Only in a last step X-ray micro diffraction data from a similar material are selectively added and re-interpreted to validate the structural conclusions derived from morphological data. Since there is no re-analysis of the X-ray micro-diffraction data, the title seems a bit misleading in this regard. In addition the re-interpretation of the X-ray micro-diffraction data is not sound and not compatible with the general conclusions drawn in this manuscript. Even more, the presented X-ray micro-diffraction data draw into question the general model of melt crystallization of banded polymer spherulites presented in this manuscript.

However, the microscopy data presented by the authors are of high quality and look very well done. The etching technique applied to reveal structures below the amorphous cover layer, or to reveal the shape of lamellar sections across the thickness of quasi-two-dimensional polymer spherulites works very well and allows a much better appreciation of the lamellar features as compared to no-etched SEM data. Therefore, I think that the microscopy data have their own merit and deserve to be published. However, drawing structural conclusions from purely morphological data is something that is only valid if the correlation between the morphological features and the underlying structure is well established. Though, in the case presented here a spherulitic crystallization model is proposed that is not established in the general community and therefore requires to be supported with structural information. I understand that the authors try to do exactly this in this manuscript, but as mentioned above the re-interpretation of the X-ray micro-diffraction data, in particular their correlation to the morphological SEM data, is contradicting.

Since the main conclusions in this manuscript is drawn from the interpretation of cross sections of lamellar objects with the surface of the spherulite or a cut/fracture surface, I would like to point out that the visible features are only of two-dimensional nature and it is not always evident to conclude on the underlying three dimensional shape. Analyzing the visible cross section of a lamellar or lamellar stack protruding the surface of the spherulite to conclude on the underlying shape is not new. I would like to point the authors attention to a publication from 1989 by A. Lustiger et al. (J Polym Sci, B: Polym Phys, 27, (1989) p. 56), featured in the review from B. Lotz and S. Cheng in POLYMER in 2005 (cited in this manuscript) where the authors elaborate on the cross section which a lamellar helicoid would generate when cut by the surface of the spherulite under an angle. Interestingly, the section shape a helicoid can reveal under such condition is very similar to the shapes reported by the authors of this manuscript.

As a result of the shape interpretation, based on the microscopic data, the authors propose a crystallization model for polymer spherulites where radially grown lamellar crystals (small dimension, typically associated with the polymer crystallographic c-axis, is perpendicular to the radial direction of the spherulite) is intermitted with sections where the lamellar platelets are stacked in the radial direction forming tangential platelets (see Figure 6 of the manuscript). If this would be true several features needs to be observable.

First, the radial growth rate of the spherulites would have to be strongly periodic and in phase with the band formation during crystallization. The radial growth rate would clearly decrease in the moment where the fast crystal growth direction deviates from the radial direction of the spherulite, while it should increase when growing in the radial direction. It would be great if the authors could provide such data for the presented polymer material.

Secondly, this would imply that the crystallographic c-axis is oriented radially in these tangential crystal stacks. This can be assessed using scanning X-ray micro diffraction or any other micro focused method giving direct or indirect information on the crystal structure.

Thirdly, the signature of the tangential platelet stacks, e.g the Lamellar correlation peak in the small angular region of the X-ray micro beam scattering, should be oriented radially and periodically in phase with the band spacing.

In the multitude of publications on banding and twisting in polymers addressed with micro-XRD, signatures compatible with the latter two points have not been reported to my knowledge. It would be very interesting if the authors could provide some references or cases where such structural signatures are observed. An alternative to micro beam XRD would be for example focused polarized infrared-spectroscopy measurements that can provide information on the orientation of molecular segment conformations as done by W. Kossak and F. Kremer published in Colloid and Polymer Science 297(3) 2019.

In general, I would say that the spherulite structure model, put forward by the authors, is not supported by the presented X-ray micro-diffraction data. Also, from the microscopy data presented in the manuscript, I actually see very nice evidence for the “classical model” of lamellar twisting and banding in spherulites.

A good example is Figure 3a in the manuscript. The authors may correct me, if I misinterpret the image. Unfortunately, the direction of growth/radial direction of the spherulite is not indicated in this beautiful SEM picture. However, from my perspective the center of the spherulite is in the top left corner of the image within the little ring shaped depression while the ridges form concentric rings around the center. This means that the straight cut through the polymer film is tangential (with respect to the concentric ridges) at the left part of the cut (looking at the growth front), while it is clearly oblique with respect to the growth direction on the right side of the image. On the left side of the image, where the radial/growth direction points towards the reader, the lamellae are edge on with the lamellae stacked tangentially, as expected from the classical twisting model for the edge-on situation. In the middle of the image where we view oblique to the growth direction, we see “c-shapes” in the valleys representative of partially flat-on crystals during the twist, as predicted by Lustiger et al., getting more and more flat as the cutting angle gets more shallow towards the right side of the image. However, I cannot identify the features of the image in the model drawn in Figure 3b. This might be due to the missing indication of the radial/growth direction in the schematics.

The same is true for Figure 4a. As I understand it, the reader looks onto the growth front and we see only edge on (ridge, lamellar stacking is tangentially) and flat on (valley, lamellar stacking is vertically) lamellae. All these shapes are expected when sectioning helicoidal stacks under the shown angles. However, again, the growth/radial axis is not indicated in the schematics in Figure 4b. So I have difficulty to identify the drawn features in the SEM image.

Concerning the data presented in Figure 5 I have to admit that I’m not able to identify any lamellar structure, neither radially nor tangentially, which is probably attributed to the fact that this is an un-etched spherulite.

From the presented images so far, I’m not really convinced of the spherulitic model presented in Figure 6, but rather see the “classical” model supported by the microscopy images as explained above.

When it comes to the micro diffraction data, and particularly the schematic drawings, shown in the manuscript in Figure 7 and 8, I’m a little bit confused. Again, in most of the drawings, and particularly in Figure 7a and 7b, the radial and growth direction is not indicated. Therefore, I will shortly describe what I see in the drawings. In Figure 7a, the spherulitic bands are clearly visible in the optical microscopy image and the location of the scanning points are indicated. The scanning direction is not exactly radially (vertically down bypassing the center of the spherulite, as I derived from the original publication), but more or less in this part of the scan. In the “classical” view, and as it was concluded in the original publication by Hsieh Y.-T. et al., the crystallographic a-axis is oriented radially. In the case of the drawing in Figure 7a the a-direction, points towards the reader (red arrow). Due to the fact, that the lattice is orthorhombic the b- and c- vectors rotate around the a-axis in the b*/c*-plane, which is normal to the a-axis (and a*-vector for orthorhombic unit cells). In the classical model this is described as helicoidal twist and is the case for polymers that exhibits lattices of high symmetry which have often a mirror or screw axis symmetry operation in the growth axis direction. This is the case for all the orthorhombic cases that have shown twisting so far as I’m aware of (e.g. PE, PLLA, PDLA, PHB, PHV, PCL, PVDF and probably more). In Figure 7a1 the authors claim that this is the case for the model with the “tangential” lamellar stacks (described in the model in Figure 6), however, in this case, as said before, the crystallographic c-axis has to be radially oriented with respect to the spherulite while the crystallographic a- and b-vectors have to be tangentially at the location where the tangential lamella are present (e.g. point f16). This is not compatible with the model presented in 7a. The only interpretation where this would be possible is if the radial direction of the spherulite is left/righ in Figure 7a. However, in this case the 020 reflection, marking the crystallographic b-direction, has to be radially oriented in the case of flat-on crystals. Looking into Figure 8d, this is never the case. Therefore I cannot follow this correlation. In my opinion, the drawing in figure 7a is well compatible with the “classical” model for a helicoidal twist (a-axis radial and fast growth axis; b and c rotate in a plane normal to/ a).

To my knowledge, the model shown in figure 7b has never been proposed or identified as crystal twisting model. It would be great if the authors could provide some background or references for this assumption as this would only be the case if the 111 crystallographic axis would be the radial direction in the spherulite (I assume that the authors mean that the 111 reciprocal vector is normal to the rotation plane, since 100 (a*), 010 (b*) and 001 (c*) can by definition not be in one plane together).  I’m not aware of such particular case. In most known cases, the fast growth axis falls together with a main crystallographic direction either a or b, but never c. In some cases for non-banded spherulites the 110 was identified as radial direction. I remember that there is one case reported for melt crystallized polymer spherulites, where the radial/rotation axis was not coinciding with a crystallographic vector while the fast crystallization direction is aligned with a crystallographic vector. In this case, of a particular  poly(propylene adipate), the lamellar ribbons formed a true helical feature (Rosenthal M. et al. Macromol Rapid Comm 34, 23-24 (2013). However this is a particular case and a case for a helical twist.

Looking at the X-ray micro-diffraction data in Figure 8d: From the original paper (Hsieh Y-T et al. POLYMER 55 (2014) 6906-6914) the direction of scanning is almost radial (as stated above), at least for the patterns indicated here. In addition the radial direction is marked by a black arrow in every XRD pattern. In the all shown patterns the 020 peak, and therefore the b-direction, is always tangentially oriented with respect to the spherulite radius when in Bragg-condition, which is the case when the crystals are inclined by the Bragg-angle towards the incoming X-ray beam, which is close to normal to the spherulite film for this d-spacing. If the spherulite model, put forward by the authors, would be true, the 020 crystallographic reflection must appear radially at least at some point in the measurement. However, in none of the displayed micro-diffraction patterns presented here or in the original publication, this is the case.

Again, from the micro-XRD, the 020 does never appear radially as it clearly should in the “valley” situation, but always tangentially. However, in Figure 9 the authors indicate a radial orientation of the b-direction. This in in clear contradiction with the micro-XRD data shown in Figure 8d. When looking in the original publication by Hsieh Y-T et al., the data indicate a rotation of the lattice around the radial crystallographic a-axis (rotation in the b*/c*-plane) as concluded in the original publication. This means that the micro-XRD data presented here are not compatible with the proposed model of the authors.

When looking at Figure 10 where the authors try to imagine how the diffraction patterns of helicoidally twisted crystals (in the idealized case of a helicoidal twist the axis of rotation in in the center of the rotating cross section, while it is moved along the axis in contrast to a helical twist, where the axis of rotation can be outside of the object) might appear, I would like to refer the authors to a publication by V.A. Luchnikov et al. (Appl Cryst 42, 4, (2009), 673-680) where a computer simulation was conducted to simulate the interactions of a X-ray micro-beam with a crystalline helicoidally twisted object with different cross sections. While the appearance of the meridional (along the rotation axis) and equatorial (perpendicular to the rotation direction) diffraction peaks only reveal a slight angular spread (that is correlated with the width of the helicoid cross section) the off meridional crystallographic reflections reveal more complicated drop-like shapes and can appear to move along the azimuth while sliding along the rotation axis. However, in no case the equatorial SAXS or WAXS reflections reveal a splitting. This can be easily understood as the central part of the helicoid contributes to the diffraction signal as well at lower spread angles forming a slightly extended arc where the arc length is correlated to the twist angle (ratio of pitch vs. width of the helicoid).

In general, I would like conclude, that the manuscript should not be published in the state it is currently in. While the body of microscopy investigations conducted is rich and very beautiful, and in addition gives beautiful inside into the morphology of the lamellar cross section in banded polymer spherulites, I do not think that the conclusions drawn from this data are justified and in clear contradiction to the presented X-ray micro-diffraction data. 

However, in order to alleviate my concerns and to support the proposed spherulite structure model, the authors could include additional information derived from various experimental and theoretical approaches. It would, for example, be nice to have information on radial growth by optical microscopy to show a modulation in the radial growth rate, as it could be expected for a model that proposes periodical tangential crystals. Also, scanning micro-SAXS data or information on the orientation of molecular segment orientation could be a way to prove the proposed spherulitic model beyond morphological information from microscopy, especially in the points where the tangential crystals are expected. Selected area electron diffraction data could also help to support the case. From a more theoretical perspective, the authors could add computer simulation/calculation showing intersection models of the different proposed lamellar shapes with respect to the cutting angles as done by Lustiger et al. for helicoidal objects. However, without such additional experimental and theoretical support the concluded structural model for PHBV/PTA spherulites is not justified.

 On page 11 of the manuscript the authors state that similar to the X-shaped WAXS pattern collected for the DNA, which revealed the double helical structure of the molecular packing, is transferable and applicable to the X-ray scattering pattern of helical and helicoidal objects as part of the morphology of semi crystalline polymers. The authors assert that the double helical structure of DNA is in part similar to that of a crystalline lamellar helicoids and, therefore, the diffraction pattern collected from helicoidal crystal ribbons should show a similar X-shape intensity distribution for the diffraction peaks in the pattern as the DNA WAXS is showing. The absence of such X-shaped pattern in the presented X-ray micro-diffraction data, presented in Figure 8, is therefore a proof that the underlying amellar morphology cannot be of helicoidal or helical nature. However, this assumption is not correct and the equation of the WAXS pattern of DNA with WAXS pattern of helicoidal and helical lamellar crystals does not stand up to scrutiny and shows a certain misunderstanding on how helical structures appear in X-ray scattering at different length scales.

In the DNA helical structure, probed in the famous X-ray photograph, the array of atoms, which are organized across multiple pitches of the molecular DNA double helix, are in crystallographic register with d-spacings in the Ångström range, which is the length scale that can be probed with wide angle X-ray diffraction. The fact that the famous X-shaped pattern can be observed is due to helical molecular arrangement on the atomic scale where atoms across several pitches can serve as diffraction grating for the coherent X-ray beam to diffract. The periodic electron density modulations then gives rise to the particular position and intensity distribution of individual diffraction peaks that appear as an X-shape in the diffraction pattern. However, in order that such diffraction can occur, the coherence length of the X-ray beam, which cannot be larger than the X-ray beam itself, has to be larger than the unit-cell of diffraction grating, which corresponds to the helical pitch of the molecule.

In contrast, the hypothetical periodicities of the helicoidal structures discussed here, are in the order of a few micrometers and up to several hundred of micrometers. The corresponding q-vector (q = 2п/d = 0.0012 nm-1, for a 5μm d-spacing), is beyond what any SAXS camera can probe. In addition, and in order to be detectable with X-ray scattering, the electron density of the object would need to be periodically modulated at these length scales and would need to be probed with an X-ray beam that has a coherence length in the order of the spacing. This means that such structures could not be observed with X-ray micro-beams that are smaller than the pitch of the object. In addition, the density of the material, and thus, the electron density, is averaged out over such large length scales due to the packing of lamellar stacks filling up the intermediate space between the pitches.

Therefore, no periodic pattern can be observed on such large length scales with X-ray scattering.

What the X-ray micro diffraction is probing in the shown scanning experiments is the local orientation of the crystallographic lattice within the lamellar ribbon, but not the pitch of the ribbon itself. The twisting period, however, can be calculated by measuring how much the X-ray beam has to travel radially outwards to reveal half the rotation of the crystallographic lattice. The twisting of these crystals will result in an increase of the width of the peaks in the q-space, since the non- planarity of the object will decrease the effective coherence length of the crystal. The periodical “non-presence” of various diffraction peaks, when scanning along the spherulites radius, comes from the fact that the Bragg-condition has to be fulfilled for a particular d-spacing in the lattice to be able to experience positive interference, which will happen twice for any crystallographic reflection of the crystallographic main axes for a full crystal rotation, but can be more often for mixed reflections with projection onto the rotation plane. For example the h11 peaks will appear at 4 times during a full crystal rotation for high symmetry lattices such as orthorhombic lattices.

Author Response

reply attached

Reviewer 2 Report

The manuscript reports one nice method to study the PHBV periodic crystals, which was tailored by crystallizing in presence of PVAc or PTA for displaying desired periodicity patterns. The co-assembled alternate-layered lamellae of banded PHBV crystal aggregates, resembling the structures in the nature’s mineral moonstone or nacre. These results are interesting, which merits publication in this journal after considering the following possible revisions.

1)     How to etch the specimens by using solvent? How to make sure that all PVAc or PTA are eliminated completely?

2)      The size in Figure 1 should be unified.

3)     It is interesting to obtain regular pattern of anti-clockwise spiraling rings. How to control the direction of the crystalline morphologies?

Author Response

reply attached

Reviewer 3 Report

see attached file

Author Response

reply attached

Reviewer 4 Report

21.076.2023

A review to evaluate its suitability for publication Type of manuscript:

The Manuscript Title: Microbeam X-ray Re-analysis on Periodically Assembled Poly(β-hydroxybutyric acid-co-β-hydroxyvaleric acid) Tailored with Diluents

Authors: Chun-Ning Wu, Selvaraj Nagarajan*, Li-Ting Lee, Chean-Cheng Su, Eamor M. Woo

The Paper presented by the Chun-Ning Wu and co-authors in Polymer Physics and Theory Section, Journal Polymers focused on one of the most important aims of the Journal - understanding of new physical phenomena (iii) and harnessing of self-assembly and biological strategies for producing complex multifunctional structures (v).

The object of this study was periodic crystals of poly(b-hydroxybutyric acid- 13 co-b-hydroxyvaleric acid) (PHBV). The authors conducted a particularly thorough study of the morphology of PHBV self-assembly in the presence of poly(vinyl acetate) (PVAc) or poly(trimethylene adipate) (PTA). This approach allowed the authors to study the mechanism of formation of periodicity patterns, crystal aggregates and resembling the structures of PHBV. For this purpose, such methods of microstructure analysis as Atomic-force microscopy, Polarized-light optical microscope, Field-Emission Scanning Electron Microscopy were applied. The results of the present study will be of practical importance in studying the periodic assembly of a wide variety of polymers with similar aggregates. In the Introduction section the aspects of formation of various structures of Periodic-aggregated spherulites are discussed in detail in accordance with numerous reviews by other authors.  

The Results and discussion section is presented with data on the formation of Periodic assembly: top surface vs. interio and Microbeam WAXD analysis for lamellae in ring-banded PHBVs. Kinetic and morphometric aspects of the growth of PHBV/PTA crystals isolated from polymer melts at different Tmax are considered. The authors successfully presented microphotographs and schematic for assemblies on top surface correlating with fractured interior lamellae of PHBV-RBS crystallized from PHBV/PTA in different ratios at Tc= 80 oC. Also very interesting are the results of displaying the interior lamellae's orientations at various spots along with the unit-cell rotation in the interior lamellae of changing inclinations with respect to the substrate surface. All this allows us to investigate in detail and present a model of self-assembly of PHBV into periodically banded aggregates.

The alternate grating periodicity in the banded PHBV can be a mimicry of many nature's structural crystals to function as photonic units for interference.

There are some questions, the answers to which will allow the authors to improve the text of the paper and recommend it for publication in Polymers Journal:

1. Can PHBV be described as a hypothetical chemical structural formula? If possible, the structure should be presented in Section 2.1. 

2.  What information do the data on PDI values for the polymer systems PHBV, PHBV/PTA, and PHBV/PVAc carry? How rational is it to present PDI values without zeta potential values, whereas both values could successfully characterize the stability of the analyzed disperse systems.

3. What is the effect of accumulation of crystal defects and impurities on their radial or tangential distribution?

4. Minor design inaccuracies occur in the manuscript text, such as in figure captions: font, color…

All these comments do not diminish the importance of the main goal and objectives of the submitted manuscript.

After minor revisions, the manuscript can be approved for publication in the Polymer Physics and Theory Section, Journal Polymers.

Respectfully, reviewer

Author Response

Reviwer-4

Round 2

Reviewer 1 Report

Please see document attached

Author Response

Reviewer
